# Expanding Horizons in Cholangiocarcinoma: Emerging Targets Beyond FGFR2 and IDH1

**DOI:** 10.3390/ijms262110755

**Published:** 2025-11-05

**Authors:** Lily Darman, Quinn Kaurich, Md Sazzad Hassan, Urs von Holzen, Niranjan Awasthi

**Affiliations:** 1Department of Chemistry and Biochemistry, University of Notre Dame, Notre Dame, IN 46617, USA; lily.a.darman@gmail.com; 2Department of Surgery, Indiana University School of Medicine, South Bend, IN 46617, USA; qkaurich@iu.edu (Q.K.);; 3Harper Cancer Research Institute, University of Notre Dame, Notre Dame, IN 46617, USA; 4Goshen Center for Cancer Care, Goshen, IN 46526, USA; 5School of Medicine, University of Basel, CH-4056 Basel, Switzerland

**Keywords:** cholangiocarcinoma, KRAS, targeted therapies, FGFR, IDH

## Abstract

Cholangiocarcinoma (CCA) is a biliary tract cancer that accounts for approximately 3% of all gastrointestinal cancers. CCA is a “silent” disease that remains undetected for a long period of time, often presenting at an advanced stage with minimal treatment options and a poor prognosis. Advanced CCA remains largely inoperable, and combination gemcitabine plus cisplatin (GemCis) chemotherapy remains the standard treatment for patients affected by this disease. There is a desperate need for new therapeutic alternatives, and extensive research is ongoing to address this gap. Targeted therapies represent a rapidly expanding area of cancer treatment and are currently under active investigation in CCA. The FDA has approved the targeted therapies ivosidenib, pemigatinib, infigratinib, and futibatinib, as well as the immunotherapy durvalumab, for patients with CCA in recent years. Several other therapeutic strategies are still under investigation, targeting molecular pathways including p53/MDM2, JAK/STAT, KRAS, HER2, VEGFR, PDGFR, MET, ALK, MAPK, PI3K/AKT, BRAF, and DNA damage repair signaling. While several promising advancements have been made, further research is required to improve outcomes for patients with CCA. This review provides an up-to-date, comprehensive overview of currently approved targeted therapies in CCA, as well as those under investigation.

## 1. Introduction

Cholangiocarcinoma (CCA) is a biliary tract cancer characterized by the formation of malignant tumors within the biliary tree [1,2]. It is the second most common primary liver cancer [3,4,5] and accounts for approximately 3% of all gastrointestinal cancers [6]. The global incidence of CCA is relatively low in Western countries, at less than 2 per 100,000 people, but is markedly higher in Thailand and other Eastern countries, where rates exceed 6 per 100,000 people [4,6]. Although several risk factors are known, its predominance in the East is largely attributed to liver fluke infections caused by *Opisthorchis viverrini* and *Clonorchis sinensis*, associated with poor sanitation and the consumption of raw fish in traditional dishes [7,8,9].

CCA is classified as intrahepatic (iCCA, 10–20%), perihilar (pCCA, 50–60%), or distal (dCCA, 10–20%) based on tumor location within the biliary tree [1,2,3,10,11,12]. iCCA arises above the second-order bile ducts, pCCA occurs between the second-order bile ducts and the junction of the cystic duct and common bile duct, and dCCA arises below this junction (Figure 1) [3,10]. Perihilar and distal CCA are collectively termed extrahepatic CCA (eCCA). CCA is often referred to as a “silent” disease due to its asymptomatic nature in the early stages, leading to late diagnosis and limited treatment options [6,13]. The disease exhibits high therapeutic resistance due to factors such as a complex tumor microenvironment (TME) and diverse genetic manifestations, resulting in poor prognosis and low five-year overall survival rate of 15–30% [2]. For patients with unresectable CCA, median overall survival (mOS) is approximately one year, with a five-year survival rate not exceeding 10% [14].

Surgery remains the only curative treatment option but is often not feasible at diagnosis. Even when resection is possible, recurrence rates approach 50% within the first year [13,15,16]. A combination of gemcitabine and cisplatin (GemCis) chemotherapy is the standard of care for advanced, inoperable CCA and has been shown to extend mOS compared to best supportive care (12.8 vs. 6.1 months) [2,17,18,19,20]. The FOLFOX regimen (5-fluorouracil plus oxaliplatin) has been approved as a second-line therapy for patients who progress on GemCis, based on the ABC-06 trial, which demonstrated clinically meaningful improvements in 6-month and 12-month survival rates [21]. More recently, the TOPAZ-1 trial led to the approval of durvalumab (an anti-PD-L1 antibody) combined with GemCis as the preferred first-line treatment for advanced or metastatic CCA, improving overall survival compared to GemCis alone (mOS: 12.9 vs. 11.3 months) [22]. In parallel, the KEYNOTE-966 trial evaluated pembrolizumab (a humanized anti-PD-1 antibody) in combination with GemCis in a similar population and demonstrated a comparable survival benefit (mOS: 12.7 vs. 10.9 months) without introducing new safety concerns (NCT04003636). Together, these two pivotal phase III studies establish chemoimmunotherapy as the current standard first-line approach for advanced or metastatic CCA [23].

Given the poor prognosis associated with CCA and limited treatment options, there remains a dire need to develop new therapies for patients affected by this deadly disease. This review provides a comprehensive overview of the current state of CCA research and highlights emerging targeted therapies that may offer potential new treatment avenues.

## 2. Recent Success of Targeted Therapies in CCA: Tyrosine Kinase Inhibitors

Although chemotherapy has long been the standard treatment for unresectable or metastatic CCA, its clinical benefit is limited by toxicity and modest survival gains [24,25].

Advances in genomic profiling have revealed a spectrum of actionable molecular alterations, paving the way for precision medicine approaches. *TP53* and *KRAS* mutations are the most common [26,27,28]. In iCCA, frequent alterations include *IDH1/2* mutations (10–20%), *FGFR2* fusions/rearrangements (10–15%), *KRAS* mutations (15–25%), *TP53* mutations (15–20%), *BAP1* loss-of-function (15–20%), *ARID1A* mutations (10–15%), *PIK3CA* mutations (5–10%), *SMAD4* mutations (5–10%), *ATM* mutations (5–10%), and *BRCA1/2* alterations (5–7%) (Figure 1) [29,30]. In eCCA, frequent alterations include *TP53* mutations (40–50%), *KRAS* mutations (20–40%), *SMAD4* inactivation (15–20%), *CDKN2A/B* loss (15–25%), *ARID1A* mutations (10–15%), *ERBB2* (*HER2*) amplification/mutations (10–15%), *PIK3CA* mutations (5–10%), *BAP1* alterations (5–10%), *ATM* mutations (5–8%), and *BRCA1/2* alterations (3–5%) (Figure 1) [31,32,33]. Between the two classifications of eCCA, *KRAS* and *TP53* mutations predominate in pCCA, whereas *SMAD4* and *ERBB2* amplification/mutations (~10–15%) are more common in dCCA. These molecular distinctions underscore potential differences in tumor biology and therapeutic vulnerabilities between pCCA and dCCA.

Identification of actionable mutations, particularly *IDH1* and *FGFR2* alterations, has enabled targeted therapies to improve patient outcomes, as discussed in the following sections. Currently approved agents include the IDH1 inhibitor ivosidenib and the *FGFR2* inhibitors pemigatinib, infigratinib, and futibatinib [34]. These advances provide hope for more effective, personalized treatment strategies for CCA patients.

### 2.1. FGFR Inhibitors

*FGFR2* gene fusions occur in ~10–15% of iCCA cases, making FGFR2 a compelling therapeutic target [35,36]. Several FGFR inhibitors have demonstrated clinical efficacy in iCCA. Pemigatinib, a selective FGFR1–3 inhibitor, demonstrated efficacy in the phase I/II FIGHT-101 trial (NCT02393248) in patients with previously treated solid tumors with or without FGFR aberrations [37]. Subsequently, the phase II FIGHT-202 trial (NCT02924376) enrolled previously treated CCA patients harboring *FGFR2* gene fusions or rearrangements, other FGFR aberrations, or no FGFR aberrations. This trial showed notable efficacy in patients with *FGFR2* fusions or rearrangements, with a median progression-free survival (mPFS) of 6.9 months and mOS of 21.1 months. Based on these findings, the FDA approved pemigatinib in 2020 for previously treated, unresectable, advanced, or metastatic CCA patients harboring *FGFR2* fusions or other alterations. Final results from the FIGHT-202 trial, published in June 2024, showed an mPFS of 7.0 months (95% CI: 6.1–10.5) and an mOS of 17.5 months (95% CI: 14.4–22.9) (Table 1) [38]. A Phase III study (NCT03656536) comparing pemigatinib vs. GemCis in treatment-naïve patients with unresectable or metastatic CCA harboring *FGFR2* fusions or rearrangements is currently ongoing.

Similarly, infigratinib, another FGFR1–3 inhibitor, demonstrated an mPFS of 7.3 months and mOS 12.2 months in a phase II study, leading to FDA approval in 2021 for patients with *FGFR2* fusions or alterations [39]. On 16 May 2024, FDA approval for infigratinib was withdrawn at the sponsor’s request due to the inability to complete confirmatory studies (https://www.fda.gov/drugs/resources-information-approved-drugs/withdrawn-fda-grants-accelerated-approval-infigratinib-metastatic-cholangiocarcinoma (accessed on 23 October 2025)) (Table 1). More recently, futibatinib, an irreversible FGFR1–4 inhibitor, demonstrated an mPFS of 8.9 months and mOS of 20.0 months in a phase II study and received FDA approval in 2022 for iCCA patients with *FGFR2* fusions or other alterations (Table 1) [40].

Beyond direct oncogenic signaling, activated FGFR pathways contribute to an immunosuppressive TME by upregulating immune checkpoint molecules such as PD-L1, suppressing T-cell function, and promoting tumor-associated macrophage polarization. These findings provide a strong rationale for evaluating FGFR inhibitors in combination with immune checkpoint inhibitors (ICIs) in FGFR-altered cancers, including CCA [41]. A phase I/II trial investigating the combination of pemigatinib plus pembrolizumab demonstrated tolerability and preliminary antitumor activity in patients with advanced malignancies, including those with FGF/FGFR alterations [42].

**Table 1 ijms-26-10755-t001:** List of FDA-approved small-molecule targeted therapies for cholangiocarcinoma (CCA), organized by molecular target and clinical indication.

Drug	Approval Year	Target	Patients	Efficacy Results	Most Common Grade ≥ 3 AEs	Trial No./Reference
Pemigatinib	2020	FGFR1, FGFR2, FGFR3 Little to no activity against FGFR4	Previously treated, advanced or metastatic CCA harboring FGFR2 fusions or rearrangements	mPFS: 7.0 mo mOS: 17.5 mo	Hyperphosphatemia 14.3% Fatigue 5% Stomatitis 7% Arthralgia 6%	NCT02924376 [38]
Infigratinib	2021 Withdrawn 2024	Selective for FGFR1–3	Previously treated, unresectable locally advanced or metastatic CCA with FGFR2 fusion/rearrangement	mPFS: 7.3 mo mOS: 12.2 mo	Hypophosphatemia 14.1% Hyperphosphatemia 12.7% Hyponatremia 11.3% Stomatitis 9.9%	NCT02150967 [39]
Futibatinib	2022	Selective for FGFR1–4	Previously treated, unresectable locally advanced or metastatic iCCA with FGFR2 fusion/rearrangement	mPFS: 9.0 mo mOS: 21.7 mo	Hyperphosphatemia 30% Elevated AST level 7% Stomatitis 6% Fatigue 6%	NCT04093362 [40]
Ivosidenib	2021	Mutant IDH1 enzyme	Previously treated, locally advanced/metastatic CCA with an IDH1 mutation	mPFS: 2.7 months mOS: 10.3 months	Anemia 7% Increased blood bilirubin 6% Hyponatremia 6% Ascites 9%	NCT02989857 [43]

### 2.2. IDH1 Inhibitors

Mutations in *IDH1/2* are present in 10–30% of iCCA cases and are less common in eCCA (~7%). *IDH1* mutations are predominant, representing 90% of *IDH* mutations, with the remainder being *IDH2* mutations [44,45,46]. Ivosidenib, an inhibitor specific to mutant *IDH1*, suppresses production of the oncometabolite 2-hydroxyglutarate, exhibiting antitumor effects. In a phase III randomized trial, ivosidenib significantly improved PFS (2.7 vs. 1.4 months) and OS (10.3 vs. 7.5 months) compared with placebo in patients with previously treated, locally advanced, or metastatic *IDH1*-mutant CCA [43]. Common treatment-related adverse events (AEs) included nausea, ascites, anemia, hyperbilirubinemia, and hyponatremia. Based on these findings, the FDA approved ivosidenib in 2021 for adult patients with *IDH1*-mutant CCA (Table 1) [43]. Although data on resistance mechanisms are limited [47], in vitro studies suggest potential resistance through isoform switching between mutant *IDH1* and *IDH2* [48].

Considering that *IDH1* mutations can promote an immunosuppressive TME, there is potential for synergy between IDH1 inhibition and ICIs [49]. A phase II trial (NCT04056910) of ivosidenib plus nivolumab (a fully human anti-PD-1 antibody) in patients with an advanced or refractory *IDH1*-mutant solid tumor, including CCA, demonstrated that all four patients with iCCA had stable disease (SD) [50]. Additionally, a phase I/II study (NCT05921760) is ongoing to assess the safety, tolerability, and preliminary efficacy of ivosidenib in combination with nivolumab and ipilimumab in previously treated patients with unresectable or metastatic *IDH1*-mutant CCA [51].

Overall, the cumulative evidence from phase II and III trials supports FGFR2 and IDH1 inhibitors as the most validated molecularly targeted approaches in advanced CCA. These therapies offer meaningful clinical benefit in biomarker-selected patients, marking a paradigm shift toward precision medicine. However, expanding these benefits beyond select genomic subgroups will require continued investigation into resistance mechanisms and combination strategies.

## 3. Immunotherapy in CCA

Immunotherapy has emerged as a promising approach in CCA, though its success has been limited by the unique features of the TME. In CCA, the TME comprises various immune cells such as B lymphocytes, cytotoxic CD8+ T cells, regulatory T cells (Tregs), natural killer (NK) cells, and tumor-associated macrophages [52]. Recent advances in ICIs, particularly those targeting the PD-1/PD-L1 axis, have shown potential in improving clinical outcomes for CCA patients, although their effectiveness as monotherapies is limited by a suppressive immune microenvironment [53].

Several studies have explored ICIs as second-line therapy for advanced CCAs (Table 2). The phase II KEYNOTE-158 trial evaluated pembrolizumab monotherapy in patients with advanced or metastatic solid tumors, including CCA, who had microsatellite instability-high (MSI-H) or mismatch repair-deficient (dMMR) tumors and had failed prior therapy. The study reported an objective response rate (ORR) of 5.8%, mPFS of 2.0 months, and mOS of 7.4 months [54]. Similarly, nivolumab monotherapy in a phase II trial showed an ORR of 10.9%, mPFS of 3.7 months, and mOS of 14.2 months, with PD-L1 positivity correlating with improved outcomes [55]. In the CA209-538 phase II trial, nivolumab plus ipilimumab (a fully human anti-CTLA-4 antibody) achieved an ORR of 23.1% and disease control rate (DCR) of 44%, with mPFS of 2.9 months and OS of 5.7 months [56]. A phase II trial (NCT03704480) assessing durvalumab (a human anti-PD-L1 antibody) plus tremelimumab (a human anti-CTLA-4 antibody) in patients who had progressed after platinum-based chemotherapy reported an ORR of 9.7%, DCR of 40.8%, mPFS of 2.5 months, and mOS of 8.0 months [57].

The most significant success in utilizing immunotherapy for CCAs has come from combining ICIs with first-line chemotherapy (Table 2). Preclinical studies suggest that the GemCis regimen enhances immunogenicity within the TME, providing a rationale for adding ICIs to this treatment [60]. A phase II BilT-01 trial evaluated GemCis with nivolumab and a separate regimen of nivolumab plus ipilimumab in advanced/metastatic CCA. The mPFS and mOS were 6.6 months and 10.6 months, respectively, in the chemoimmunotherapy arm, compared to 3.9 months and 8.2 months, respectively, in the dual ICI arm. Neither regimen significantly exceeded the historical control PFS threshold, indicating that adding nivolumab to standard chemotherapy did not provide a clear benefit, and the nivolumab-ipilimumab combination was less effective [58]. Another phase II trial (NCT03046862) investigated GemCis followed by GemCis plus durvalumab and tremelimumab; GemCis plus durvalumab; and GemCis plus durvalumab and tremelimumab. The ORRs were 50.0, 72.3, and 70.2%; mPFS were 6.6, 11.8 and 12.3 months; and mOS were 15.0, 20.2 and 18.7 months, respectively. These findings suggested that initiating immunotherapy early in treatment was beneficial, while the addition of dual ICI therapy provided minimal additional benefit compared to monotherapy [59].

The most notable success in integrating ICIs into first-line treatment was demonstrated in the phase III TOPAZ-1 trial, where GemCis plus durvalumab significantly improved outcomes over GemCis (ORR: 26.7 vs. 18.7%; mPFS: 7.2 vs. 5.7 months; mOS: 12.9 vs. 11.3 months), leading to FDA approval of the GemCis plus durvalumab regimen in September 2022. The incidence of grade 3 or 4 adverse events was similar between the durvalumab (75.7%) and placebo (77.8%) groups (Table 2) [22]. Similarly, the KEYNOTE-966 trial showed improved outcomes with pembrolizumab combined with GemCis compared with chemotherapy alone (mPFS: 6.5 vs. 5.6 months; mOS: 12.7 vs. 10.9 months) (Table 2) [23].

Ongoing first-line trials continue to explore various ICIs combined with chemotherapy. A phase III trial (NCT03478488) is investigating the efficacy of the subcutaneously administered anti-PD-L1 antibody envafolimab in combination with gemcitabine/oxaliplatin (GEMOX) [61]. Several phase II trials are also underway, including NCT04191343 (toripalimab plus GEMOX), NCT03796429 (toripalimab plus gemcitabine and S-1), and NCT04172402 (nivolumab plus gemcitabine and S-1).

Although immune checkpoint blockade—particularly PD-1/PD-L1 inhibitors—has improved outcomes for some patients, its benefits remain limited by an immunosuppressive TME. To expand therapeutic options beyond immune-based approaches and established FGFR2 and IDH1 inhibitors, current research is exploring additional molecular targets that drive CCA progression. The following section highlights these emerging investigational targeted therapies and their potential to reshape treatment paradigms.

## 4. Investigational Targeted Therapies in CCA

Building on the positive outcomes and approval of small-molecule inhibitors targeting FGFR and IDH oncogenic signaling in CCA, several other therapeutic strategies focusing on different oncogenic pathways are currently being explored for advanced-stage CCA. Some of the most promising signaling pathways with therapeutic potential include the following:

### 4.1. P53-MDM-2 Signaling

MDM2-p53 antagonists have primarily been investigated in hematologic malignancies. However, agents such as idasanutlin (a selective MDM2 antagonist) and ALRN-6924 (a dual MDM2/MDMX inhibitor) have demonstrated tolerability and promising efficacy in patients with advanced solid tumors [62,63]. These antagonists are also being evaluated in combination with ICIs, as they may promote a pro-immunogenic TME and exhibit synergy with anti-PD-1 antibodies in preclinical models [64].

Milademetan (RAIN-32), an MDM2-p53 antagonist, has been assessed in solid tumors, including CCA, with phase I trials showing manageable safety profiles and modest antitumor activity [65]. In a Phase II study, milademetan was evaluated in patients with advanced or metastatic solid tumors, including CCA, whose tumors harbored *MDM2* gene amplification and wild-type *p53*. In CCA patients, the treatment demonstrated approximately 29% tumor regression. The ORR across all tumor types was 19.4% (Appendix A) [66]. Currently, brigimadlin (BI 907828), another potent MDM2-p53 antagonist, is under development for CCA patients (Table 3). This compound has demonstrated encouraging antitumor effects in preclinical models of dedifferentiated liposarcoma with *MDM2* amplification [67] and showed synergistic efficacy when combined with a PD-1 checkpoint inhibitor in a colon cancer mouse model [68].

Preliminary data from phase Ia/Ib trials evaluating brigimadlin, either as monotherapy or in combination with the PD-1 inhibitor ezabenlimab, in CCA patients have been encouraging. Among treated patients, six achieved a partial response (PR), and four had SD. Responses were durable, and no unexpected toxicities were reported, though some patients experienced grade 3/4 treatment-related AEs such as neutropenia, thrombocytopenia, and anemia [69]. Based on these promising results, a phase IIa/IIb trial of brigimadlin is currently recruiting patients with advanced CCA, pancreatic ductal adenocarcinoma (PDAC), or other solid tumors (NCT05512377) [87].

### 4.2. ErbB Inhibitors

The ErbB family consists of receptor tyrosine kinases ErbB1 (EGFR), ErbB2 (HER2), ErbB3, and ErbB4 [88]. Epidermal growth factor receptor (EGFR) and HER2 activate various pathways that stimulate cell proliferation and differentiation [89]. In iCCA, approximately 10–32% of cases exhibit EGFR and/or HER2/neu overexpression [88,90,91].

For EGFR-targeted therapies, a randomized phase III trial investigated the addition of the EGFR inhibitor erlotinib to GEMOX chemotherapy. GEMOX and GEMOX + erlotinib resulted in ORRs of 15.8 vs. 29.6%, mPFS of 4.2 vs. 5.8 months, and mOS of 9.5 months in both groups, indicating no OS benefit with the addition of erlotinib (NCT01149122) (Table 3) [70]. A phase II study of the EGFR monoclonal antibody cetuximab in combination with GEMOX showed promising antitumor activity (ORR: 23%) (NCT01216345) [71]. However, another phase II study assessing GEMOX with and without cetuximab reported an improvement in mPFS but no significant difference in mOS between treatment arms (NCT01267344) (Table 3) [72]. Other ErbB inhibitors, such as the EGFR inhibitor panitumumab and the dual EGFR/HER2 inhibitor lapatinib, failed to show significant clinical benefit in biliary tract cancer patients (Appendix A) [92,93].

For HER2-targeted therapies, a retrospective study found that the HER2 monoclonal antibody trastuzumab provided clinical benefit in gallbladder cancer with HER2/neu overexpression but not in CCA [94]. In a phase II study, the HER2 inhibitor vandetanib failed to increase PFS in CCA patients (NCT00753675) (Appendix A) [95]. In the phase II SUMMIT basket trial (NCT01953926), the pan-HER inhibitor neratinib demonstrated modest activity in HER2-mutant biliary tract cancers, with PFS and OS outcomes comparable to standard therapies, suggesting limited but potential therapeutic relevance in selected patients (Table 3) [73].

### 4.3. VEGFR and PDGFR Inhibitors

Targeting the tyrosine kinase receptors VEGF and PDGF may provide therapeutic benefit for CCA patients, as these RTKs are commonly overexpressed [91,96]. In a phase II study, the VEGFR inhibitor bevacizumab combined with gemcitabine-capecitabine failed to show significant clinical improvement (mPFS: 8.1 months, mOS: 10.2 months) in CCA patients (Appendix A) [97]. However, an earlier phase II study combining bevacizumab with GEMOX demonstrated antitumor activity in CCA patients (mPFS: 7.0 months; mOS: 12.7 months) (Table 3) [74].

Sorafenib, a multikinase inhibitor targeting VEGFR, PDGFR, and Raf, has been extensively evaluated in clinical trials for CCA but has failed to produce significant benefits, either as monotherapy or in combination. In a phase II trial, sorafenib monotherapy was evaluated in patients with advanced CCA and showed modest efficacy, reporting an mPFS of 2.3 months and an mOS of 4.4 months (Table 3) [75]. The SWOG 0514 phase II trial involving chemotherapy-naïve patients with unresectable or metastatic gallbladder carcinoma or CCA, single-agent sorafenib yielded no confirmed objective responses, an mPFS of ~3.0 months, and an mOS of ~9.0 months (Appendix A) [98]. The SWOG phase II trial investigated the combination of sorafenib and erlotinib in advanced gallbladder carcinoma or CCA. The regimen demonstrated limited activity, with an mPFS of 2 months and an mOS of 6 months (Table 3) [76]. Ramucirumab, a VEGFR2 antibody, showed promising results in a single-arm phase II trial with PFS comparable to that of current chemotherapy regimens (mPFS: 3.2 months; mOS: 9.5 months) (Table 3) [77]. Another promising VEGF inhibitor regorafenib (an oral multikinase inhibitor of VEGFR, PDGFR, and RAF kinases), in a phase II study of chemotherapy refractory advanced CCA patients, demonstrated mPFS of 3.6 months and mOS of 7.3 months (NCT02053376), highlighting its potential as a future treatment for patients with advanced CCA (Table 3) [78].

### 4.4. MET Inhibitors

Overexpression of MET, another RTK, is associated with poor prognosis in CCA and is detected in approximately 35–50% of iCCAs, making it a promising therapeutic target [89,99]. However, clinical trials evaluating MET inhibitors have yielded mixed results (Table 3).

Cabozantinib, a dual MET/VEGFR2 inhibitor, was assessed in a phase II clinical trial for CCA patients but exhibited significant toxicity and failed to demonstrate meaningful antitumor activity (mPFS: 1.8 months, mOS: 5.2 months) (Appendix A) [100]. A phase II trial of ramucirumab plus the MET inhibitor merestinib with GemCis showed tolerability but no improvement in mPFS in CCA patients (Appendix A) [101]. The MET inhibitor tivantinib in combination with gemcitabine, demonstrated low toxicity and potential antitumor activity in a phase I trial of advanced CCA patients [102]. Tepotinib, a selective MET inhibitor, showed promising clinical activity in the phase II VISION trial involving NSCLC patients with *MET* exon 14 mutations (NCT02864992) [103]. A long-term follow-up of the VISION trial reaffirmed tepotinib’s efficacy in this patient population [104]. Following its FDA approval, tepotinib may represent a viable treatment option for CCA patients harboring *MET* mutations [105].

### 4.5. ALK Inhibitors

The receptor tyrosine kinases *ALK* and *ROS1* are altered in approximately 1.1–8.8% of iCCA cases [89,106]. Due to structural similarities between ALK and ROS1 proteins, ALK inhibitors have been investigated as potential therapies for *ROS1*-mutant CCA patients [89]. In a phase III trial in NSCLC patients with *ALK* mutations, the ALK inhibitor ceritinib significantly improved PFS compared to chemotherapy, suggesting potential efficacy for *ROS1*-mutant CCA patients as well [107]. Ceritinib was investigated in CCA patients with ROS1 or ALK overexpression (NCT02374489), but the trial was terminated due to insufficient patient recruitment. Alectinib and crizotinib, two other ALK inhibitors, have been evaluated in *ALK*-mutated non-lung solid tumors [108]. Results revealed that alectinib, and possibly crizotinib, may benefit these patients, suggesting potential therapeutic relevance for CCA patients with *ROS1* mutations [108]. Entrectinib, an ALK and ROS1 inhibitor, was tested in two phase I clinical trials in patients with solid tumors, including gastrointestinal tract tumors. These trials demonstrated that entrectinib was well tolerated and exhibited promising antitumor activity across multiple tumor types [109]. Entrectinib is currently being investigated in a phase II basket study in solid tumors, including CCA harboring *ROS1*, *ALK*, or *NTRK1/2/3* mutations (NCT02568267).

### 4.6. MAPK Signaling Inhibitors

Oncogenic activation of the RAS/RAF/MEK/ERK (MAPK) signaling pathway is driven by specific mutations in the *RAS* genes, leading to constitutive activation of RAS proteins and subsequent uncontrolled proliferation, differentiation, migration, and metastasis [110]. MAPK pathway components are well-known oncogenic drivers in many cancers, making them attractive targets for therapy [110,111]. *BRAF* mutations are found in only 3.3% of iCCA cases [112]. Vemurafenib, a BRAF V600-specific inhibitor, was tested in eight CCA patients with *BRAF V600* mutation, but only one partial response was observed [113]. However, a case study reported complete response in a CCA patient with *BRAF V600* mutation treated with vemurafenib, panitumumab, and irinotecan [114]. A next-generation BRAF inhibitor PLX8394 has also been evaluated in *BRAF*-mutant solid tumors, including CCA. A phase II trial evaluated selumetinib, a MEK1/2 inhibitor, in patients with metastatic CCA who had received prior chemotherapy, demonstrated limited clinical activity, with a mPFS of 3.7 months and a mOS of 9.8 months, highlighting modest benefit in the second-line setting (Table 3) [79]. A phase 1b trial of selumetinib combination with GemCis in chemotherapy-naïve advanced CCA patients demonstrated promising activity, with an mPFS of 6.4 months [115]. However, a randomized phase II study of selumetinib plus GemCis failed to improve efficacy and increased toxicity in advanced CCA patients compared to GemCis (mPFS: 6.0–7.0 vs. 6.3 months; mOS: 11.7 vs. 12.8 months) (Appendix A) [116]. Another MEK inhibitor, trametinib, was tested in a phase II trial in Japan in CCA patients, demonstrating SD in 65% of patients and progressive disease (PD) in 35%, with mPFS of 10.6 weeks and ~20% 1-year OS (Table 3) [80]. A phase II trial comparing trametinib to chemotherapy (5-FU or capecitabine) in refractory CCA patients failed to demonstrate any benefit of trametinib over chemotherapy (ORR: 0 vs. 5%; mPFS: 1.5 vs. 3.3 months; mOS: 4.3 vs. 6.6 months) (Appendix A) [117]. A combination of trametinib and the VEGF receptor inhibitor pazopanib in advanced CCA patients who were refractory to or refused standard-of-care treatment options showed modest activity, with an ORR of 5%, DCR of 75%, mPFS of 3.6 months, and mOS of 6.4 months. These results suggest that while the combination therapy demonstrated some clinical activity, the overall efficacy was limited and it did not result in a statistically significant improvement in PFS (Table 3) [81]. By contrast, in a phase II trial of dabrafenib (a BRAF inhibitor) plus trametinib in *BRAF V600*-mutant CCA patients showed promising activity with low toxicity (ORR: 47%; mPFS: 7.2 months; mOS: 11.3 months) (Table 3) [82]. A phase II trial of atezolizumab (anti-PD-L1 antibody) with or without the MEK inhibitor cobimetinib in CCA patients demonstrated improved PFS with the combination (3.65 vs. 1.87 months). However, the low response rate in both arms highlighted the immune-resistant nature of CCA (Table 3) [83]. Similarly, a randomized phase II study evaluated atezolizumab plus the CD27 immune agonist varlilumab (CDX-1127), with or without cobimetinib, and was terminated early due to lack of clinical benefit (ORR: 0 vs. 4%; mPFS: 2.2 vs. 1.8 months) (Appendix A) [118].

### 4.7. PI3K/AKT/mTOR Signaling Inhibitors

The PI3K/AKT/mTOR pathway is a key regulator of cell growth and survival, frequently dysregulated in CCA. Mutations in *PI3K* occur in approximately 6% of eCCA and 4% of intrahepatic iCCA tumors, while *PTEN* loss or mutation is detected in 4–5% of cases [112]. Overactivation of phosphorylated AKT is observed in the majority of CCA samples, underscoring the pathway’s therapeutic relevance [119,120].

The mTOR inhibitor everolimus demonstrated modest activity (mPFS: 3.2 months; mOS: 7.7 months) in patients with previously treated CCA in a phase II trial [84]. Another phase II trial evaluating everolimus monotherapy as first-line treatment in advanced biliary tract cancer reported a mPFS of 5.5 months and mOS of 9.5 months but failed to meet its primary endpoint (Appendix A) [121]. The PI3K inhibitor copanlisib plus GemCis achieved an ORR of 17.4% [122], while the AKT inhibitor MK2206 showed no clinical activity (ORR: 0%; mPFS: 1.7 months; mOS: 3.5 months) in a small trial of eight patients (Appendix A) [123].

Overall, while preclinical rationale is strong, clinical benefits from PI3K/AKT/mTOR inhibitors in CCA remain limited, highlighting the need for better patient selection and combination strategies.

### 4.8. JAK/STAT Pathway Inhibitors

The JAK/STAT pathway, particularly STAT3, regulates cell proliferation, differentiation, and apoptosis and is frequently activated in CCA through cytokine (e.g., IL-6) and RTK signaling [124,125,126]. Persistent STAT3 activation promotes tumor growth, and elevated STAT3 expression has been documented in iCCA tissues [127]. AZD9150, a novel targeted drug, has shown efficacy in reducing STAT3 expression in various preclinical cancer models [128,129], and is currently being evaluated in pancreatic cancer (NCT02983578) [128]. Other STAT3 inhibitors, such as OPB-31121 for hepatocellular carcinoma (NCT01406574) and BBI608 for colorectal cancer (NCT01830621), are under clinical investigation [128]. While mechanistically promising, JAK/STAT inhibitors remain in early stages of evaluation for CCA, warranting further preclinical and translational research.

### 4.9. PARP Inhibitors

*BRCA1*-associated protein 1 (BAP1) functions as a tumor suppressor that regulates cell cycle progression, differentiation, cell death, and DNA damage response. Loss-of-function *BAP1* mutations sensitize cancer cells to PARP inhibitors (PARPi), similar to *BRCA*-mutated cancers [130]. In CCA, *BAP1* mutations occur exclusively in iCCA patients (~15–20%) [29,30], though some series have reported frequencies as high as 35% [131].

A case report described a refractory metastatic CCA patient with a novel *BAP1* mutation who experienced a prolonged response to the PARP inhibitor olaparib (>11 months) [132]. In a retrospective study in iCCA patients with *IDH1* mutation who had previously received at least one line of cisplatin-based therapy, PARPi monotherapy showed a PFS ranging from 1.4 to 18.5 months and OS from 2.8 to 42.4 months [133]. Several trials are currently investigating PARPi in combination with ICIs in advanced CCA. The phase II BilT-02 trial evaluating rucaparib (a PARPi) plus nivolumab as maintenance therapy after first-line platinum-based therapy did not meet its primary endpoint (PFS at four months) but demonstrated a high DCR of 77%, with a preliminary mPFS of 4.6 months and OS of 15.9 months (Table 3) [85]. A phase II trial of olaparib (a PARPi) plus pembrolizumab in advanced CCA patients demonstrated acceptable safety but did not significantly improve outcomes (ORR: 15.4%; mPFS: 5.45 months; mOS: 7.2 months) (Table 3) [86]. Similarly, a phase II study of olaparib and durvalumab in *IDH*-mutated CCA was discontinued after stage I due to lack of efficacy (no complete or partial responses; mPFS: 1.97 months) (Appendix A) [134]. However, a case report described a complete response to olaparib plus pembrolizumab after platinum-based induction chemotherapy in an iCCA patient with a *BRCA2* mutation and PD-L1-positive recurrent iCCA [135].

Although biologically compelling, PARPi use in CCA remains exploratory, and future studies should refine patient selection based on molecular profiles such as BAP1 or BRCA alterations.

## 5. Potential of Targeting *KRAS* in CCA

Gain-of-function mutations in the *KRAS* gene are among the most common genetic alterations in CCA, occurring in approximately 15–25% of iCCA cases and up to ~40% of pCCA and dCCA cases [94,136,137]. These mutations lead to constitutive activation of the MAPK and PI3K/AKT/mTOR signaling pathways, promoting cancer cell proliferation and survival. *KRAS* mutations are associated with a more aggressive tumor phenotype and reduced survival rates in CCA [138], highlighting the potential of mutant *KRAS* as a therapeutic target.

Historically, direct targeting of KRAS has been challenging. However, the development of *KRAS^G12C^*-specific inhibitors, such as sotorasib, has provided a new avenue for therapy [139]. The prevalence of different *KRAS* missense mutations varies across cancer types. For instance, *KRAS^G12C^* mutations are rare in pancreatic cancer (2–3% of all *KRAS* mutations) and CCA (8%) but represent the predominant *KRAS* mutation subtype in NSCLC (39–42%) [140]. As a result, KRAS inhibitors like sotorasib and adagrasib, which specifically target the *KRAS^G12C^* mutation, are more effective in subsets of lung adenocarcinoma rather than CCA. In CCA, the most prevalent *KRAS* mutations are *G12D* (~48%), *G12V* (~18%), and *G12S* (~12%) [141]. Therefore, the development of mutant allele-specific inhibitors for *KRAS^G12D^*, *KRAS^G12V^* or *KRAS^G12S^*, or broad-spectrum pan-KRAS inhibitors, holds significant therapeutic promise for CCA.

Several KRAS inhibitors are currently being evaluated in clinical trials for tumors harboring *KRAS* mutations. In a phase II study (NCT03785249) of adagrasib in patients with advanced solid tumors carrying the *KRAS^G12C^* mutation, a subgroup of 8 CCA patients demonstrated an ORR of 50.0%, an mPFS of 11.3 months, and an mOS of 15.1 months (Table 4) [142]. Moreover, an exploratory analysis from the TOPAZ-1 trial, which evaluated chemotherapy plus immunotherapy, found that *KRAS*-mutated CCAs were more frequently observed among long-term survivors who lived more than 18 months after randomization [143]. An ongoing phase I study (NCT06607185) is assessing the safety and tolerability of LY4066434, a pan-KRAS inhibitor, in patients with *KRAS*-mutant advanced or metastatic solid tumors, including CCA (Table 4). In addition, a phase I/II study of the *KRAS^G12D^* inhibitor MRTX1133 was initiated in solid tumors including CCA with *KRAS^G12D^* mutation, but the trial was terminated early before proceeding to phase II (NCT05737706) (Table 4).

Overall, investigational targeted therapies for CCA encompass a diverse range of pathways, including p53/MDM2, ErbB, VEGFR/PDGFR, MET, ALK, MAPK, PI3K/AKT/mTOR, JAK/STAT, PARP, and KRAS signaling. While preclinical and early-phase clinical data show encouraging activity, most of these agents remain exploratory, with limited evidence from randomized trials. Notably, recent advances in the direct targeting of KRAS using pan-KRAS inhibitors and isoform-selective inhibitors represent a significant milestone, expanding therapeutic potential beyond previously validated FGFR2 and IDH1 pathways. These developments, together with emerging combination approaches and biomarker-driven trial designs, highlight a rapidly evolving landscape that may redefine precision treatment paradigms in CCA.

## 6. Comparative Stratification of Therapeutic Modalities

Given the expanding therapeutic landscape in CCA, a comparative understanding of efficacy, safety, and patient selection is essential. Among approved treatments, FGFR2 inhibitors (pemigatinib, futibatinib, infigratinib) and the IDH1 inhibitor ivosidenib provide the greatest benefit in molecularly defined subgroups with manageable toxicity. Chemoimmunotherapy (GemCis plus durvalumab) remains the preferred first-line option for patients without targetable alterations, offering the best overall survival improvement in unselected populations. Investigational agents, including KRAS, MDM2, and PARP inhibitors, show emerging promise but await phase III validation.

Treatment selection should be guided by comprehensive genomic profiling, prior therapy, and individual patient factors. In practice, *FGFR2* or *IDH1* alterations should prompt use of targeted therapy, whereas chemoimmunotherapy remains the standard backbone for biomarker-negative disease. Rational combinations integrating targeted agents with immunotherapy or chemotherapy may further enhance efficacy and overcome resistance.

Predictive clinico-genomic parameters are increasingly critical for refining patient selection. *FGFR2* fusions and *IDH1* mutations predict response to targeted inhibitors; MSI-H/dMMR status, PD-L1 positivity, and high tumor mutational burden correlate with improved immunotherapy outcomes; and *BAP1* or *BRCA1/2* alterations indicate sensitivity to PARP inhibition. Incorporating these biomarkers into routine diagnostics will be key to achieving truly personalized treatment in CCA.

## 7. Conclusions and Future Perspective

CCA remains a highly aggressive malignancy with limited therapeutic options and poor overall prognosis. In recent years, the emergence of molecularly targeted therapies has provided new opportunities to improve clinical outcomes. The discovery of actionable genetic alterations has already been translated into approved therapies, including the IDH1 inhibitor ivosidenib and the FGFR2 inhibitors pemigatinib, infigratinib, and futibatinib, which represent the first steps toward precision medicine in CCA. Moreover, the integration of immunotherapy into the treatment paradigm, highlighted by the approval of durvalumab in combination with chemotherapy, underscores the potential of harnessing the immune system against this disease.

Beyond these established approaches, several other molecular targets are being actively investigated, such as the p53/MDM2 axis, VEGFR, MET, ALK, MAPK, PI3K/AKT/mTOR, JAK/STAT, and BAP1-related pathways. *KRAS*, long considered an “undruggable” oncogene, is now a particularly promising target due to recent advances in the development of selective *KRAS* inhibitors. Continued exploration of these and other pathways holds the potential to expand the therapeutic armamentarium against CCA.

The future of CCA management lies in the integration of comprehensive genomic profiling, biomarker-driven clinical trials, and rational drug combinations that overcome resistance. Incorporating targeted, immunotherapeutic, and chemotherapeutic strategies, guided by tumor heterogeneity and molecular signatures, will be key to achieving durable responses. Adaptive and platform-based trial designs will further accelerate evaluation of emerging therapies and enable real-time learning.

Translating these advances into clinical practice remains challenging. Barriers such as limited access to next-generation sequencing, inconsistent biomarker testing, high treatment costs, and disparities in healthcare infrastructure hinder global implementation of precision oncology. Addressing these limitations through collaborative research networks and data sharing will be essential to ensure equitable access to emerging therapies.

Moving forward, CCA treatment should prioritize molecular stratification at diagnosis, combination regimens that jointly target oncogenic signaling and immune evasion, and adaptive trial platforms that rapidly incorporate novel biomarkers. Such an integrated approach will optimize treatment selection, accelerate development of durable personalized therapies, and ultimately improve survival and quality of life for patients with this devastating disease.

## Figures and Tables

**Figure 1 ijms-26-10755-f001:**
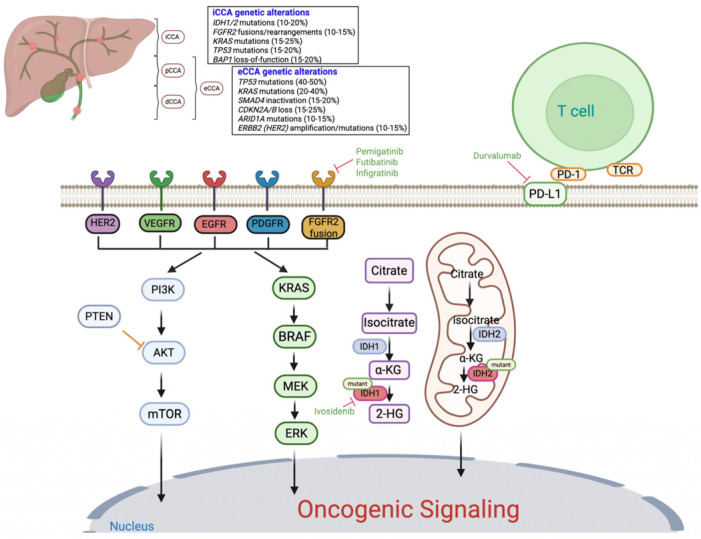
Anatomical and molecular landscape of cholangiocarcinoma and key oncogenic signaling pathways: The figure illustrates the anatomical classification of cholangiocarcinoma (CCA) into intrahepatic (iCCA), perihilar (pCCA), and distal (dCCA) subtypes, along with representative genetic alterations commonly observed in each. Major oncogenic signaling cascades implicated in CCA pathogenesis are shown, including the RAS/MAPK and PI3K/AKT/mTOR pathways, which regulate cell proliferation, apoptosis, and survival. Tyrosine kinase receptor alterations (e.g., *FGFR2*, *HER2*, *EGFR*, *VEGFR*, *PDGFR*) and downstream mutations (*KRAS*, *BRAF*, *PTEN*) contribute to aberrant signaling. Mutations in *IDH1* and *IDH2* lead to accumulation of the oncometabolite 2-hydroxyglutarate (2HG), promoting epigenetic dysregulation. Approved targeted therapies are indicated in green text, including FGFR inhibitors (pemigatinib, futibatinib, infigratinib), the IDH1 inhibitor (ivosidenib), and immune checkpoint blockade (durvalumab, targeting PD-L1). Additional potential therapeutic targets with investigational agents currently under evaluation are also shown.

**Table 2 ijms-26-10755-t002:** Summary of Phase II and III clinical trials evaluating immunotherapy, either as monotherapy or in combination with chemotherapy, for the treatment of cholangiocarcinoma (CCA).

Treatment	Phase	Line	Patients	Efficacy Results	Most Common Grade ≥ 3 AEs	Trial No./Reference
Pembrolizumab	II	2nd	Adults with advanced/metastatic solid tumors (including CCA), MSI-H/dMMR tumors, prior therapy failure	ORR: 5.8% mPFS: 2.0 mo mOS: 7.4 mo	Common all grade AEs: Fatigue 14.4%, rash 11.5%, pruritus 8.7%	NCT02628067 [54]
Nivolumab	II	2nd	Adults with advanced/metastatic CCA, refractory to ≥1 prior therapy	ORR: 11% mPFS: 3.7 mo mOS: 14.2 mo	Hyponatremia 6% Increased alkaline phosphatase 4%	NCT02829918 [55]
Ipilimumab/Nivolumab	II	2nd	Adults with advanced/metastatic CCA, refractory to ≥1 prior therapy	ORR: 23.1% mPFS: 2.9 mo mOS: 5.7 mo	Immune-related (any grade) 49% Immune-related (≥3) 15%	NCT02923934 [56]
Durvalumab/Tremelimumab (interim)	II	2nd	Adults with advanced/metastatic CCA, who failed prior platinum-based chemotherapy	ORR: 9.7% mPFS: 2.5 mo mOS: 8.0 mo	Common any grade AE: Fatigue, pruritus, increased liver enzymes	NCT03704480 [57]
GemCis + nivolumab Nivolumab + Ipilimumab	II	1st	Adults with advanced/unresectable CCA	ORR: 22.9 vs. 3.0% mPFS: 6.6 vs. 3.9 mo mOS: 10.6 vs. 8.2 mo	Arm A: Neutropenia 34%, anemia 23%, fatigue 8.6%; Arm B: Elevated transaminases 9.1%	NCT03101566 [58]
GemCis, then GemCis + durvalumab + tremelimumab GemCis + durvalumab GemCis + durvalumab + tremelimumab	II	1st	Adults with previously untreated advanced CCA	ORR: 50 vs. 72.3 vs. 70.2% mPFS: 6.6 vs. 11.8 vs. 12.3 mo mOS: 15 vs. 20.2 vs. 18.7 mo	Across all patients: Neutropenia 53% Anemia 40% Thrombocytopenia 19%	NCT03046862 [59]
GemCis + durvalumab GemCis	III	1st	Adults with previously untreated advanced (unresectable or metastatic) CCA, or recurrent disease ≥6 months post-surgery/adjuvant therapy	ORR: 26.7 vs. 18.7% mPFS: 7.2 vs. 5.7 mo mOS: 12.9 vs. 11.3 mo	Neutropenia 21 vs. 25% Anemia 19 vs. 19% Thrombocytopenia 10 vs. 11% Fatigue 6 vs. 5%	NCT03875235 [22]
GemCis + pembrolizumab GemCis	III	1st	Adults with previously untreated, advanced/metastatic/unresectable CCA	ORR: 28.7 vs. 28.5% mPFS: 6.5 vs. 5.6 mo mOS: 12.7 vs. 10.9 mo	Neutropenia 21 vs. 25% Anemia 19 vs. 19% Thrombocytopenia 10 vs. 11% Fatigue 6 vs. 5%	NCT04003636 [23]

**Table 3 ijms-26-10755-t003:** Selected Phase II and III clinical trials of targeted therapies demonstrating clinical efficacy or modest activity, administered as monotherapy or in combination with chemotherapy and/or immunotherapy, in cholangiocarcinoma (CCA).

Treatment	Phase	Line	Patients	Efficacy Results	Most Common Grade ≥ 3 AEs	Trial No./Reference
Brigimadlin (MDM2–p53 antagonist) Brigimadlin + Ezabenlimab	II	2nd	Advanced or metastatic, MDM2-amplified and p53-wt solid tumors including CCA	ORR: 33 vs. 67%	Neutropenia 26 vs. 25% Thrombocytopenia 24 vs. 22% Anemia 15 vs. 14%	NCT03449381 NCT03964233 [69]
GEMOX GEMOX + Erlotinib (EGFR inhibitor)	III	1st	Treatment-naïve patients with unresectable locally advanced or metastatic CCA	ORR: 15.8 vs. 29.6% mPFS: 4.2 vs. 5.8 mo mOS: 9.5 vs. 9.5 mo	Neutropenia 6 vs. 4% Thrombocytopenia 4 vs. 4% Peripheral neuropathy 3% (GEMOX) Diarrhea 7% (GEMOX + Erlotinib)	NCT01149122 [70]
GEMOX + Cetuximab (EGFR inhibitor)	II	1st	Treatment-naïve, unresectable advanced or metastatic CCA	ORR: 23%	Skin rash 13.3% Peripheral neuropathy 13.3% Thrombocytopenia 10%	NCT01216345 [71]
GEMOX GEMOX + Cetuximab	II	1st	Treatment-naïve patients with unresectable locally advanced or metastatic CCA	ORR: 15 vs. 27% mPFS: 4.1 vs. 6.7 mo mOS: No difference	Neutropenia 6.7 vs. 11.3% Skin rash 0 vs. 6.5% Allergic/infusion reaction 0 vs. 4.8%	NCT01267344 [72]
Neratinib (HER2 inhibitor)	II	2nd	Advanced/metastatic CCA, HER2-mutant, refractory	ORR: 16% mPFS: 2.8 mo mOS: 5.4 mo	Diarrhea	NCT01953926 [73]
GEMOX + Bevacizumab	II	1st	Advanced, measurable CCA ≤1 prior chemo	mPFS: 7.0 mo mOS: 12.7 mo	Neutropenia 20% Sepsis 9.4% Thrombocytopenia 8.6%	NCT00361231 [74]
Sorafenib (VEGFR, PDGFR, Raf inhibitor)	II	2nd	Advanced CCA who progressed after prior chemotherapy	ORR: 2% mPFS: 2.3 mo mOS: 4.4 mo	Skin rash 15% Gastrointestinal toxicity 15% Hand-foot skin reaction 11%	[75]
Sorafenib + Erlotinib	II	1st or 2nd	Unresectable or metastatic disease and could have received 0–1 prior systemic chemotherapy	mPFS: 2 mo mOS: 6 mo	Hypertension Elevated liver enzymes Diarrhea	[76]
Ramucirumab (VEGFR2 inhibitor)	II	2nd or later	Advanced, unresectable CCA; prior gemcitabine-based treatment	ORR: 1.7% mPFS: 3.2 mo mOS: 9.5 mo	Hypertension Proteinuria Pulmonary embolism	[77]
Regorafenib (VEGFR2 inhibitor)	II	2nd or later	Advanced or metastatic CCA, progression after first-line chemotherapy.	ORR: 10.7% mPFS: 3.6 mo mOS: 7.3 mo	Hypophosphatemia 40% Hyperbilirubinemia 26% Hypertension 23%	NCT02053376 [78]
Selumetinib (MEK1/2 inhibitor)	II	2nd or later	Metastatic CCA	ORR: 12%; SD: 44% mPFS: 3.7 mo mOS: 9.8 mo	Diarrhea Nausea Fatigue	[79]
Trametinib (MEK1/2 inhibitor)	II	2nd	Advanced or metastatic CCA	SD: 65%; PD: 35% mPFS: 10.6 wks 1-year OS: 20%	Rash, Diarrhea Fatigue Elevated liver enzymes	NCT01943864 [80]
Pazopanib (VEGFR inhibitor) + Trametinib	II	2nd or later	Advanced, unresectable CCA	ORR: 5% mPFS: 3.6 mo mOS: 6.4 mo	Thrombocytopenia 24% Rash 12% Elevated liver enzymes 12%	[81]
Dabrafenib (targets BRAF^V600E^ mutation) + Trametinib	II	2nd or later	BRAFV600E-mutated, unresectable, metastatic, locally advanced, or recurrent CCA	ORR: 47% mPFS: 7.2 mo mOS: 11.3 mo	Increase ***γ***-GGT 12% Decreased WBC 7% Hypertension 7%	NCT02034110 [82]
Atezolizumab (anti-PD-L1) + Cobimetinib (MEK inhibitor) Atezolizumab	II	2nd or later	Unresectable CCA, 1–2 prior systemic therapies	ORR: 3% both arms mPFS: 3.65 vs. 1.87 mo	Arm A: Rash 16%, thrombocytopenia 11%, elevated AST/ALT 8%; Arm B: Fatigue 5%, hypertension 5%, elevated AST/ALT 5%	NCT03201458 [83]
Everolimus monotherapy (mTOR inhibitor)	II	2nd	Advanced CCA who progressed after 1st-line chemotherapy	DCR: 44.7% (at 12 wks) mPFS: 3.2 mo mOS: 7.7 mo	Grade ≥ 3 AE not reported All grade AEs: Fatigue 43.6%, Thrombocytopenia 35.6%, Pyrexia 30.8%	EudraCT 2008-007152-94 [84]
Rucaparib (PARP inhibitor) + Nivolumab	II	2nd or later	Advanced/metastatic CCA, post ≥4–6 mo platinum-based therapy without progression	PR: 6.4%, SD: 71% mPFS: 4.6 mo mOS: 15.9 mo	Fatigue 29% Anemia 22.6% Neutropenia 19.3%	NCT03991832 [85]
Olaparib (PARP inhibitor) + Pembrolizumab	II	2nd or later	Advanced or metastatic CCA who have progressed after prior systemic therapy	ORR: 15.4% mPFS: 5.45 mo mOS: 7.2 mo	Anemia 35.7% Decreased neutrophil count 7.1% Diarrhea 7.1%	NCT04306367 [86]

**Table 4 ijms-26-10755-t004:** Ongoing clinical trials of KRAS inhibitors as monotherapy and in combination in cholangiocarcinoma (CCA).

Treatment	Phase	Line	Patients	Efficacy Results	Most Common Grade ≥ 3 AEs	Trial No./Reference
Adagrasib (*KRAS^G12C^* inhibitor) monotherapy	II	2nd or later	Advanced solid tumors including CCA harboring a *KRAS^G12C^* mutation who progressed after prior therapies	ORR: 50.0% mPFS: 11.3 mo mOS: 15.1 mo	Grade 3 AEs overall: 25.4% Common grade 3 events: Fatigue 6.3%, QT prolongation 6.3%	[142]
LY4066434 (pan-KRAS inhibitor) monotherapy	I	2nd or later	KRAS mutant advanced or metastatic solid tumors, including CCA	Not yet reported	Not yet reported	NCT06607185
MRTX 1133 (*KRAS^G12D^* inhibitor) monotherapy	I/II	2nd or later	*KRAS^G12D^* mutant advanced or metastatic solid tumors, including CCA	Not yet reported	Not yet reported	NCT05737706

## Data Availability

No new data were created or analyzed in this study. Data sharing is not applicable to this article.

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
