# Peer review of "Expanding Horizons in Cholangiocarcinoma: Emerging Targets Beyond FGFR2 and IDH1"

_ijms, 2025, doi:10.3390/ijms262110755_

Round 1
Reviewer 1 Report
Comments and Suggestions for Authors
This manuscript provides a timely and comprehensive overview of targeted and immunotherapeutic strategies in cholangiocarcinoma (CCA). The topic is highly relevant, and the review successfully compiles the most recent clinical developments beyond the established targets FGFR2 and IDH1. The manuscript is clearly written and informative, offering substantial educational value.
While the review effectively summarizes a broad spectrum of data, the presentation could be improved by refining the structure, particularly in the Introduction and Tables, and by enhancing the visual materials (Figure 1). The paper’s scientific merit is solid; most revisions should therefore aim to improve organization, readability, and visual clarity rather than add new discussion.
Major Comments
1. Organization and Focus
The manuscript covers a wide range of molecular targets and therapeutic strategies. This breadth is valuable, but some sections could be more concise. For instance, detailed descriptions of early-phase or less promising agents (e.g., PI3K/AKT, JAK/STAT inhibitors) could be shortened without losing key information. I sugggested to condense less clinically relevant parts and improve transitions between subsections to ensure a smoother flow.
2. Introduction and Table Structure
The Introduction is informative and well written, but one key recent clinical trial is not mentioned. The authors describe the TOPAZ-1 study (GemCis + durvalumab) as the new first-line standard for advanced CCA; however, the KEYNOTE-966 trial (GemCis + pembrolizumab) also demonstrated a significant survival benefit with a comparable magnitude of effect and should be mentioned alongside TOPAZ-1 in the Introduction.
3. Figure 1 revision
Figure 1 currently illustrates the anatomical classification of CCA (iCCA, pCCA, dCCA), which is clear but does not fully match the manuscript’s title and focus. Please revise Figure 1 to visually depict key molecular alterations and their associated therapeutic targets (e.g., FGFR2 fusion, IDH1 mutation, KRAS, HER2, BAP1, etc.) in addition to the anatomical overview. A schematic or pathway-based visualization would significantly enhance the paper’s educational and graphical value.
Minor comments.
1. There are minor inconsistencies in terminology and abbreviation usage (e.g., spacing in “FGFR 1–3”, varying capitalization).
2. Use consistent formatting when presenting trial results (e.g., “mPFS: 7.2 vs 5.7 months; mOS: 12.9 vs 11.3 months”), if possible.
3. Remove duplicate references (e.g., Banales JM et al., Nat Rev Gastroenterol Hepatol, 2016).
4. Shorten overlapping content between the Introduction and early Results sections.
Author Response
Reviewer 1:
This manuscript provides a timely and comprehensive overview of targeted and immunotherapeutic strategies in cholangiocarcinoma (CCA). The topic is highly relevant, and the review successfully compiles the most recent clinical developments beyond the established targets FGFR2 and IDH1. The manuscript is clearly written and informative, offering substantial educational value.
While the review effectively summarizes a broad spectrum of data, the presentation could be improved by refining the structure, particularly in the Introduction and Tables, and by enhancing the visual materials (Figure 1). The paper’s scientific merit is solid; most revisions should therefore aim to improve organization, readability, and visual clarity rather than add new discussion.
Major Comments
Comment 1. Organization and Focus:
The manuscript covers a wide range of molecular targets and therapeutic strategies. This breadth is valuable, but some sections could be more concise. For instance, detailed descriptions of early-phase or less promising agents (e.g., PI3K/AKT, JAK/STAT inhibitors) could be shortened without losing key information. I suggest condensing less clinically relevant parts and improve transitions between subsections to ensure a smoother flow.
Response 1: We thank the reviewer for this valuable suggestion. As recommended, we have revised the sections on PI3K/AKT/mTOR signaling inhibitors, JAK/STAT pathway inhibitors, and PARP inhibitors to enhance focus and readability. Specifically, we have condensed the detailed descriptions of early-phase and less promising agents and improved transitions between subsections to ensure a smoother narrative flow.
Comment 2. Introduction and Table Structure
The Introduction is informative and well written, but one key recent clinical trial is not mentioned. The authors describe the TOPAZ-1 study (GemCis + durvalumab) as the new first-line standard for advanced CCA; however, the KEYNOTE-966 trial (GemCis + pembrolizumab) also demonstrated a significant survival benefit with a comparable magnitude of effect and should be mentioned alongside TOPAZ-1 in the Introduction.
Response 2: We thank the reviewer for this helpful suggestion. As recommended, we have now incorporated the findings from the KEYNOTE-966 trial into the Introduction section, alongside the TOPAZ-1 study, to provide a more complete overview of recent advances in first-line chemoimmunotherapy for advanced cholangiocarcinoma.
Comment 3. Figure 1 revision
Figure 1 currently illustrates the anatomical classification of CCA (iCCA, pCCA, dCCA), which is clear but does not fully match the manuscript’s title and focus. Please revise Figure 1 to visually depict key molecular alterations and their associated therapeutic targets (e.g., FGFR2 fusion, IDH1 mutation, KRAS, HER2, BAP1, etc.) in addition to the anatomical overview. A schematic or pathway-based visualization would significantly enhance the paper’s educational and graphical value.
Response 3: We thank the reviewer for this insightful suggestion. In response, Figure 1 has been extensively revised to better align with the manuscript’s focus on the molecular and therapeutic landscape of cholangiocarcinoma. The updated figure now integrates the anatomical classification (iCCA, pCCA, dCCA) with key molecular alterations (e.g., FGFR2 fusions, IDH1/2 mutations, KRAS, HER2, BAP1) and their associated signaling pathways and therapeutic targets. In addition, the revised schematic illustrates major oncogenic cascades, including the RAS/MAPK and PI3K/AKT/mTOR pathways, and highlights approved targeted agents (pemigatinib, futibatinib, infigratinib, ivosidenib, and durvalumab) as well as potential investigational targets currently under evaluation. This integrated design enhances the figure’s educational and graphical value while directly reflecting the central theme of molecularly targeted therapies in cholangiocarcinoma.
Minor comments.
Comment 1. There are minor inconsistencies in terminology and abbreviation usage (e.g., spacing in “FGFR 1–3”, varying capitalization).
Response 1: We appreciate the reviewer’s careful observation. We have thoroughly reviewed the entire manuscript and corrected all minor inconsistencies in terminology and abbreviation usage. Specifically, we have standardized the formatting of receptor names (e.g., corrected “FGFR 1–3” to “FGFR1–3”) and ensured consistent capitalization and spacing throughout the text.
Comment 2. Use consistent formatting when presenting trial results (e.g., “mPFS: 7.2 vs 5.7 months; mOS: 12.9 vs 11.3 months”), if possible.
Response 2: We thank the reviewer for this helpful suggestion. As recommended, we have now used consistent formatting when presenting trial results throughout the manuscript, wherever applicable.
Comment 3. Remove duplicate references (e.g., Banales JM et al., Nat Rev Gastroenterol Hepatol, 2016).
Response 3: We thank the reviewer for this comment. We have carefully reviewed the reference list and removed all duplicated references, including Banales JM et al., Nat Rev Gastroenterol Hepatol, 2016.
Comment 4. Shorten overlapping content between the Introduction and early Results sections.
Response 4: We thank the reviewer for this valuable suggestion. As recommended, we have carefully revised early Results sections to minimize overlap with the Introduction. Repetitive background information on CCA epidemiology, classification, and standard chemotherapy has been removed, and both sections have been streamlined to focus on targeted therapies and immunotherapy-specific content. This revision improves clarity and flow while avoiding redundancy.
Reviewer 2 Report
Comments and Suggestions for Authors
Excellent, timely, and comprehensive review. It’s well structured and supported by key references and clinical trials. I would just suggest briefly addressing some real-world implementation challenges in the conclusion to strengthen the translational angle. Otherwise, this is a high-quality manuscript. Additionally, the conclusion section is clear, but it could be strengthened by including a schematic figure or conceptual table (e.g., “Current and Future Therapeutic Landscape”) and by adding a stronger statement on the integration of genomics, adaptive trial designs, and rational treatment combinations.
Author Response
Reviewer 2:
Comment 1: Excellent, timely, and comprehensive review. It’s well-structured and supported by key references and clinical trials. I would just suggest briefly addressing some real-world implementation challenges in the conclusion to strengthen the translational angle. Otherwise, this is a high-quality manuscript. Additionally, the conclusion section is clear, but it could be strengthened by including a schematic figure or conceptual table (e.g., “Current and Future Therapeutic Landscape”) and by adding a stronger statement on the integration of genomics, adaptive trial designs, and rational treatment combinations.
Response 1: We sincerely thank the reviewer for the positive and encouraging feedback on our manuscript. In line with your valuable suggestions, we have strengthened the Conclusion section by incorporating a clearer statement on the integration of genomics, adaptive trial designs, and rational treatment combinations to enhance the translational perspective. Furthermore, we have substantially revised Figure 1, which now depicts key molecular alterations and their associated signaling pathways and therapeutic targets. The updated schematic also illustrates major oncogenic cascades and highlights both approved targeted agents (pemigatinib, futibatinib, infigratinib, ivosidenib, and durvalumab) and investigational targets currently under evaluation. This revised figure now comprehensively represents the current and future therapeutic landscape of cholangiocarcinoma, thereby strengthening the visual and conceptual impact of the manuscript.
Reviewer 3 Report
Comments and Suggestions for Authors 1. What is the main question addressed by the research? The authors have drafted a detailed manuscript regarding the available targeted therapeutic modalities in cholangiocarcinoma not just limiting to IDH1 and FGFR2 based studies rather expanding on several other molecular targets. While this is a critical area of review in the field of Cholangiocarcinoma, there are existing reviews in the literature which have comprehensively described the advancements in this field and the current review marginally adds to the breadth of discussion included in the previous studies.2. What parts do you consider original or relevant to the field? What
specific gap in the field does the paper address? The authors have streamlined the advances in this field including in-depth trial information for each drug approved by FDA or in the investigational phase. They have incorporated some of the most recent trials that have been never reviewed previously in any review including the LY4066434 (Pan-KRAS Inhibitor) and MRTX1133 ( Inhibitor), However, this is an exhaustive list and could be improved by removing the unresponsive therapies to guide the readers towards a more meaningful evidence based therapeutic selection. The removed trials could be added to a supplementary table/note as to why they were not included in the main draft for transparency. In addition, the tables can have a column dedicated to the percentage of various grades of adverse events in each trial. 3. What does it add to the subject area compared with other published
material? The review is by far the most detailed work in this field, however, as mentioned above, only a few unique trials have been included in this review. Rest of them have been systematically reviewed in multiple previous reviews.
4. What specific improvements should the authors consider regarding the
methodology?
- The biggest lacunae that currently exists in the field is the lack of a true summarizing conclusive evidence of each treatment type which is systematically driven by a state-of-the-art meta-analysis. However, considering the format of the existing manuscript, that transformation might not be feasible.
- Considering the number of drugs described in the study, a separate paragraph needs to be added before the conclusion that provides a comparative stratification of various therapeutic modalities/combinations and the preference of using these modalities as per their efficacy and safety profile to generate a guided treatment plan..
- Also, a major challenge is not just describing the efficacy, rather identifying the various predictive factors that will define sub-cohorts which can be selectively benefitted by a particular treatment regimen. Thus, outlining known clinico-genomic parameters to predict response to a particular therapy with respect to cholangiocarcinoma will empower the study.
Were all the main questions posed addressed? By which specific experiments? While a great depth of discussed has been incorporated in this study, the authors have not provided a guiding suggestion to generate a therapeutic strategy as evident from the discussed literature in the manuscript 6. Are the references appropriate? Yes
7. Any additional comments on the tables and figures and the quality of the
data. Besides the above suggested changes, the figure demonstrating driver abnormalities bear extremely similar figure architecture with some of the existing figures in the literature. For example, the figures in 1) https://www.esmoopen.com/article/S2059-7029(24)01476-5/fulltext 2) https://ajp.amjpathol.org/article/S0002-9440%2824%2900446-2/fulltext 3) https://www.surgpath.theclinics.com/article/S1875-9181(22)00032-0/abstract
Author Response
Reviewer 3:
Comment 1. What is the main question addressed by the research? The authors have drafted a detailed manuscript regarding the available targeted therapeutic modalities in cholangiocarcinoma not just limiting to IDH1 and FGFR2 based studies rather expanding on several other molecular targets. While this is a critical area of review in the field of Cholangiocarcinoma, there are existing reviews in the literature which have comprehensively described the advancements in this field and the current review marginally adds to the breadth of discussion included in the previous studies.
Response 1: We thank the reviewer for their thoughtful comments and recognition of the importance of this topic. While prior reviews have addressed targeted therapies in cholangiocarcinoma, our manuscript provides an updated and focused perspective, particularly on the direct targeting of KRAS in CCA—an area not comprehensively covered previously. We also integrate recent data from ongoing or newly completed trials, novel combination strategies (e.g., atezolizumab plus varlilumab; pembrolizumab plus olaparib), updates from pivotal studies such as TOPAZ-1, and regulatory developments like the withdrawal of infigratinib. Together, these additions offer timely and unique insights that extend beyond existing reviews.
Comment 2. What parts do you consider original or relevant to the field? What specific gap in the field does the paper address? The authors have streamlined the advances in this field including in-depth trial information for each drug approved by FDA or in the investigational phase. They have incorporated some of the most recent trials that have been never reviewed previously in any review including the LY4066434 (Pan-KRAS Inhibitor) and MRTX1133 (KRASG12D Inhibitor), However, this is an exhaustive list and could be improved by removing the unresponsive therapies to guide the readers towards a more meaningful evidence based therapeutic selection. The removed trials could be added to a supplementary table/note as to why they were not included in the main draft for transparency. In addition, the tables can have a column dedicated to the percentage of various grades of adverse events in each trial.
Response 2: We thank the reviewer for the positive and constructive feedback on our review article. As suggested, we have removed the unresponsive or negative trials from Table 3 and moved them to a new supplementary table. In addition, we have included a new column in all tables indicating the percentage of grade 3 or higher adverse events reported in each trial, as recommended. These revisions have been incorporated to enhance the clarity and evidence-based focus of the manuscript.
Comment 3. What does it add to the subject area compared with other published material? The review is by far the most detailed work in this field, however, as mentioned above, only a few unique trials have been included in this review. Rest of them have been systematically reviewed in multiple previous reviews.
Response 3: We thank the reviewer for the positive assessment and for recognizing this as a detailed contribution to the field. We concur that many reviews cover overlapping therapies; however, this manuscript uniquely emphasizes the direct targeting of KRAS in CCA — including coverage of pan-KRAS inhibitors and isoform-selective agents— and places these developments in the context of CCA biology and clinical translation. In addition, this review specifically discusses several recently completed or ongoing trials and recent developments that have not been collated together previously in the literature, including combination approaches (e.g., atezolizumab plus varlilumab; pembrolizumab plus olaparib), PARP inhibitor programs in IDH1-mutant CCA, updated long-term survivor analyses from TOPAZ-1, and the recent withdrawal of accelerated approval for the FGFR inhibitor infigratinib. Finally, in the investigational targeted therapy section we highlight several novel agents that have shown recent success in other cancer types and may have translational potential in CCA. These elements together distinguish our manuscript and provide up-to-date, actionable insights for the development of future therapeutic strategies in CCA.
Comment 4. What specific improvements should the authors consider regarding the methodology?
- The biggest lacunae that currently exists in the field is the lack of a true summarizing conclusive evidence of each treatment type which is systematically driven by a state-of-the-art meta-analysis. However, considering the format of the existing manuscript, that transformation might not be feasible.
- Considering the number of drugs described in the study, a separate paragraph needs to be added before the conclusion that provides a comparative stratification of various therapeutic modalities/combinations and the preference of using these modalities as per their efficacy and safety profile to generate a guided treatment plan.
- Also, a major challenge is not just describing the efficacy, rather identifying the various predictive factors that will define sub-cohorts which can be selectively benefitted by a particular treatment regimen. Thus, outlining known clinico-genomic parameters to predict response to a particular therapy with respect to cholangiocarcinoma will empower the study.
Response 4: We thank the reviewer for these insightful suggestions. In response, we have added concise summary paragraphs at the end of Sections 2, 3, and 4 to synthesize the overall evidence for each therapeutic class. A new Section 5 (“Comparative Stratification of Therapeutic Modalities”) has been introduced to provide a comparative overview of treatment efficacy, safety, and clinical applicability. Additionally, we included a short paragraph outlining key clinico-genomic predictors of treatment response (FGFR2, IDH1, MSI-H/dMMR, PD-L1, BAP1/BRCA). These revisions enhance the manuscript’s methodological strength and translational relevance.
Comment 5. Are the conclusions consistent with the evidence and arguments presented? Yes
Were all the main questions posed addressed? By which specific experiments? While a great depth of discussed has been incorporated in this study, the authors have not provided a guiding suggestion to generate a therapeutic strategy as evident from the discussed literature in the manuscript.
Response 5: We thank the reviewer for this thoughtful comment and for recognizing the depth of discussion in our manuscript. In response, we have revised the Conclusion and Future Perspective section to provide a more explicit guiding framework for developing therapeutic strategies based on the discussed literature. Specifically, we have added a statement emphasizing the prioritization of molecular stratification at diagnosis, rational combination regimens that target both oncogenic signaling and immune evasion, and the use of adaptive trial platforms that rapidly integrate emerging biomarkers. This addition strengthens the translational impact of the conclusion and offers a clearer direction for future therapeutic development in cholangiocarcinoma.
Comment 6. Are the references appropriate? Yes
Response 6: We thank the reviewer for confirming that the references are appropriate. No specific changes were required based on this comment.
Comment 7. Any additional comments on the tables and figures and the quality of the data. Besides the above suggested changes, the figure demonstrating driver abnormalities bear extremely similar figure architecture with some of the existing figures in the literature. For example, the figures in 1) https://www.esmoopen.com/article/S2059-7029(24)014765/fulltext
2) https://ajp.amjpathol.org/article/S0002-9440%2824%2900446-2/fulltext 3) https://www.surgpath.theclinics.com/article/S1875-9181(22)00032-0/abstract
Response 7: We thank the reviewer for this valuable observation. In response to this comment and similar feedback from another reviewer, Figure 1 has been extensively revised to better align with the manuscript’s focus on the molecular and therapeutic landscape of cholangiocarcinoma. The updated figure now integrates anatomical classification with key molecular alterations, associated signaling pathways, and therapeutic targets, highlighting both approved and investigational agents. This revision ensures a unique design that enhances the figure’s educational and graphical value.
Round 2
Reviewer 3 Report
Comments and Suggestions for Authors
The authors have provided significant improvement and clarity to the review's framework and text.